# Activation Patterns of Functional Brain Network in Response to Action Observation-Induced and Non-Induced Motor Imagery of Swallowing: A Pilot Study

**DOI:** 10.3390/brainsci12101420

**Published:** 2022-10-21

**Authors:** Hao Xiong, Jin-Jin Chen, John M. Gikaro, Chen-Guang Wang, Feng Lin

**Affiliations:** 1Department of Rehabilitation Medicine, Sir Run Run Hospital Nanjing Medical University, Nanjing 211100, China; 2Department of Rehabilitation Medicine, The First Affiliated Hospital of Nanjing Medical University, Nanjing 210029, China; 3School of Rehabilitation Medicine, Nanjing Medical University, Nanjing 210029, China

**Keywords:** motor imagery, swallowing, dysphagia, MEG, functional brain networks, event-related spectral perturbations (ERSPs)

## Abstract

Action observation (AO) combined with motor imagery (MI) was verified as more effective in improving limb function than AO or MI alone, while the underlying mechanism of swallowing was ambiguous. The study aimed at exploring the efficacy of AO combined with MI in swallowing. In this study, twelve subjects performed the motor imagery of swallowing (MI-SW) during magnetoencephalography (MEG) scanning, and trials were divided into three groups: the non-induced group (control group, CG), male AO-induced group (M-AIG), and female AO-induced group (F-AIG). We used event-related spectral perturbations (ERSPs) and phase locking value (PLV) to assess the degree of activation and connectivity of the brain regions during MI-SW in the three groups. The results showed that compared to CG, F-AIG and M-AIG significantly activated more brain regions in the frontoparietal, attention, visual, and cinguloopercular systems. In addition, M-AIG significantly activated the sensorimotor cortex compared to CG and F-AIG. For the brain network, F-AIG and M-AIG increased the diffusion of non-hub hot spots and cold hubs to the bilateral hemispheres which enhanced interhemispheric functional connectivity and information transmission efficiency in the MI-SW task. This study provided supporting evidence that AO induction could enhance the effect of MI-SW and supported the application of AO-induced MI-SW in clinical rehabilitation.

## 1. Introduction

Dysphagia is a common dysfunction in neurological disorders, which commonly occurs in stroke, Parkinson’s disease, brain injury, dementia, multiple sclerosis, amyotrophic lateral sclerosis, and cerebral palsy [1,2,3]. In addition, the impairment of swallowing-related organic structures (e.g., neck and oropharyngeal cancer) and aging could lead to dysphagia [1,3]. Dysphagia is associated with nutritional and respiratory impairment, and increases the risk of aspiration, mortality, and medical expenses [1,4]. Compared to conventional rehabilitation programs of neurological disorders, motor imagery (MI) and action observation (AO), which were developed in recent years, avoided the risks of aspiration [5]. AO and MI could improve motor function and performance by activating mirror neurons, which were activated by actual movement [6]. AO and MI were established as effective in activating swallowing-related brain regions [7,8], and in improving limb motor function [9]. However, AO or MI alone sometimes did not achieve the ideal therapeutic efficacy. Romkema et al. reported that compared to subjects in the control group who received prosthetic training alone, no significant motor improvements were found in healthy subjects who received additional imagery and action observation on the basis of prosthetic training after 5 days’ training [10]. Crajé et al. also described that MI training significantly improved hand function, such as stretching and grasping of the affected side, but not the fine dexterity [11]. The inefficiency of MI might be due to the complexity of the task and the short training time. Therefore, the combination of AO and MI was proposed as a rehabilitation program by researchers. Currently, it has been proven that AO combined with MI can achieve better efficacy in limb motor function compared to MI alone [12,13]. However, there was a lack of relevant studies on swallowing tasks.

Magnetoencephalography (MEG) provided information about brain function at a high temporal and spatial resolution. In this study, we used MEG to collect motor imagery of swallowing (MI-SW) data in healthy young people under AO-induced and non-induced conditions, respectively. The aim of this pilot study was to provide preliminary validation that AO-induced swallowing imagery was more effective than swallowing imagery alone, and to provide a reference for future studies and clinical treatment of dysphagia patients. We hypothesized that consistent with previous studies on limb motor imagery, AO-induced MI-SW would activate swallowing-related regions of interest (ROIs) significantly stronger than MI-SW alone (i.e., non-induced MI-SW). In addition, AO induction would enhance interhemispheric functional connectivity during MI-SW. In the rest of this manuscript, the experimental paradigm, and methods for data collection and processing are described first. Next, the manuscript shows the differences of ROIs’ activation and functional connectivity of the groups, and we include a discussion of the results.

## 2. Material and Methods

### 2.1. Participants

Twelve healthy subjects (six males and six females) with ages ranging from 21 years to 26 years (mean = 23.17 years, standard deviation = 1.67 years) were recruited in our study, with an average number of years of education of 16.33 ± 1.60 years. The inclusion criteria were: (i) right handedness; (ii) the average score of the modified KVIQ-10 ≥ 2.5 [14] and the swallowing imagery score ≥ 3; (iii) the score of mini-mental state examination (MMSE) > 24 [15]. Exclusion criteria were: subjects with (i) visual and visuospatial disorders; (ii) hearing disorders; (iii) cognitive impairment or mental dysfunction; iv) contraindications for magnetic resonance imaging (MRI), or cannot tolerate 30 min MEG or 10 min MRI examinations. Since the currently used imagery questionnaire does not include items of swallowing imagery, and in order to test the subjects’ ability of swallowing imagery, we added visual and kinesthetic swallowing imagery items to the kinesthetic and visual imagery questionnaire-10 (KVIQ-10) [16]. The MMSE score and modified KVIQ-10 score of subjects were 29.67 ± 0.47 and 4.44 ± 0.47, respectively. The item of visual imagery of swallowing of the subjects was rated as 4.67 ± 0.62, and the item of kinesthetic imagery of swallowing was rated as 3.83 ± 0.69, where the score of the two items showed statistical significance (*p* < 0.01). All participants signed an informed consent form that was approved by the Ethics Committee of Sir Run Run Hospital, Nanjing Medical University (No.2019-SR-002). All examinations were carried out under the guidance of the Declaration of Helsinki.

### 2.2. Methods

#### 2.2.1. Data Acquisition

The MEG recordings were obtained using a whole-head CTF 275-channel MEG system (VSM Medical Technology Company, Vancouver, BC, Canada). Before data acquisition, with participants lying supine, three electromagnetic coils were attached as reference markers on the left and right pre-auricular points and the nasion of each participant to check the head position. During tasks, head movements exceeding 0.5 cm were excluded. The sampling rate of the MEG recording was 1200 Hz. Electroencephalogram (EEG) [17,18] leads were positioned over and under bilateral orbitals, outer canthus, dorsal hands, and wrists to record the electrical activity of the eyes and heart. To remove electromyogram signals induced by unconscious swallowing, leads were attached on bilateral submental and infrahyoid muscles [19].

Stimulus materials were presented using BrainX (Cincinnati Children’s Hospital Medical Center, Cincinnati, OH, USA) [20]. The images were projected on a projection screen 40 cm above the eyes, and the swallowing sounds were transmitted through earphones. After the MEG examination, vitamin E pellets were stuck to the positions identical to the landmarks. For the MEG source analysis, T1-weighted images of the brain were obtained using a 1.5 T MRI scanner from the GE company in the USA (TR: 33 ms; TE: 9 ms; recording matrix: 256 × 256 pixels; excitation: 1; the field of view: 240 mm; and slice thickness: 1.4 mm).

#### 2.2.2. Experimental Design

The experiment contained two tasks: AO-induced MI-SW and non-induced MI-SW. In total, 12 participants were asked to perform the non-induced MI-SW task first and then perform the AO-induced MI-SW task. Before the start of the experiments, subjects were asked to avoid performing actual swallowing actions during imagination [8,21]. Additionally, they were required to use kinesthetic imagery and first-person perspective imagery strategies of swallowing as vividly as possible to improve the quality of MI-SW [22]. Each task consisted of two blocks, with a 5 min rest period between the blocks, and each block comprised 20 trials. Each trial lasted for 19 s and contained the following procedures.

In the non-induced MI-SW task, a 6 s black-background image with grey “−” was presented first, and subjects were asked to watch the image and rest. After that, a black-background picture with “ready to start” in Chinese was displayed as a cue for 3 s. Then, a 10 s black-background image with grey “+” was displayed on the screen, in which subjects were asked to perform swallowing imagery of themselves consecutively at a comfortable pace. Furthermore, the imagination needed to be stopped when the next black-background picture with a grey “−” emerged, and so on repeatedly until the end of the experiment (Figure 1).

In the AO-induced MI-SW task, a 3 s(s) swallowing video as the cue material was displayed for each trial to induce MI-SW instead of the “ready to start” picture. The rest of the parts of the trial stayed the same as the non-induced MI-SW task. We used 10 young adults’ (age from 20 to 30 years, 5 males and 5 females) swallowing videos from the front and side view as stimulus materials for the AO-induced MI-SW task (totally including 10 males’ and 10 females’ swallowing videos). The videos were taken from the jaw to the clavicle and did not include facial features. Each video lasted for 3 s, and the action of laryngeal elevation and swallowing sounds were displayed at the 1500-milliseconds (ms) time point. The swallowing videos were displayed randomly by sampling without replacement, and each video was presented only once in each block (Figure 1). In this research, the presence of the grey image with “+” (i.e., the beginning of MI-SW) was set as the zero point. The baseline was set as time windows of −5~−3 s to remove visual signal interferences during imaging. In addition, the cue materials of all groups were played at −3~0 s. Finally, ROIs’ activation and the network analysis were performed only for the time window of the MI-SW process (0~10 s).

Previous studies reported that different stimulus videos affect the activation of the mirror neuron system [23]. Furthermore, previous studies on the audiovisual stimulus of swallowing used males’ swallowing videos, but they did not state the reason for not using females’ swallowing videos [7,24,25,26]. We speculated that there were differences between the induction effect of different genders’ swallowing videos. Thus, in this study, the trials of two tasks were divided into the non-induced group (control group, CG) and AO-induced group. In addition, the AO-induced group was further divided into two groups (i.e., male AO-induced MI-SW group and female AO-induced MI-SW group) according to watching males’ or females’ videos of swallowing. Eventually, the experiment included 80 effective trials, of which 20 trials were from the male AO-induced group (M-AIG), 20 trials were from the female AO-induced group (F-AIG), and 40 trials were from the control group (CG). After the experiment, all subjects reported that they had tried their best to complete the MI-SW tasks during the experiment.

#### 2.2.3. Data Processing

BrianSuite18a was used to read MRI data, and the brain cortex was annotated as 130 brain regions according to USCBrain Atlas [27]. Then, the anatomical data with USCBrain atlas and MEG files were imported to Brainstorm [28]. We established a 3D coordinate system of the Montreal Neurological Institute in the light of markers and fusing the annotated individual brain. Data were preprocessed by DC-offset correction, linear trend removal, band pass filter of 1~40 Hz, 50 Hz notch filter, and bad channel removal. Artifacts were removed by signal-space projection (SSP) and an independent component analysis (ICA) [29]. In addition, the data derived from every trial were source reconstructed by applying structural imaging data and the minimum norm imaging method, and projected to the standard template of the USCBrain-BrainSuite-2017 to generate a comparable source file.

#### 2.2.4. ROI Activation Indicators

For non-phase locked MI-SW signals, a time-frequency analysis was used to calculate event-related spectral perturbations (ERSPs) in the mission [19,21,30]. The power of the baseline window was E_b_, and the power of the task window was E_t_. ERSPs = (E_t_ − E_b_)/E_b_. ERSPs > 0 were called event-related synchronization (ERS), and ERSPs < 0 were called event-related desynchronization (ERD). Due to the band specificity of ERSPs, it was easily observed only at ~10 Hz and ~20 Hz [31], and in other frequency bands (e.g., delta, theta, and gamma), it tended to show only ERS patterns [32]. In addition, most studies of motor imagery focused on the alpha and beta bands [21,33]. Therefore, in the present study we only analyzed the alpha and beta bands.

The projected source files were used for the Morlet wavelet time-frequency transformation to calculate ERSPs as activation indicators for ROIs [34]. ERSPs for alpha (8–13 Hz) and beta (14–29 Hz) bands in the MI-SW (0~10 s) task time window were calculated. In addition, 130 brain regions were parcellated into 8 function systems [35]: ① frontoparietal system; ② attention system; ③ motor and somatosensory system; ④ cinguloopercular system; ⑤ visual system; ⑥medial default mode system; ⑦ ventral temporal association system; ⑧ auditory system.

#### 2.2.5. Construction and Analysis of Brain Network

Undirected weighted networks of different groups in different frequency bands were constructed by Pajek5.08 [36] and VOS viewer 1.6.13 [37]. The brain network was constructed and analyzed with the graph theory method, using 130 brain regions as nodes and PLV values as edge weight. Theoretically, there were coupling relationships in all brain regions, and the set of a threshold for intercepting strong couplings between brain regions during MI-SW was essential. The threshold interception method in this study was modified from a method described by Gonuguntla et al. [28] which considered three groups:$$Threshold=2\times \frac{\left\{\left[PLV_{M-MIG}^{ERSPs}\right]+\left[PLV_{F-MIG}^{ERSPs}\right]+\left[PLV_{CG}^{ERSPs}\right]\right\}}{3}$$
where $\left[PLV_{M-MIG}^{ERSPs}\right]$, $\left[PLV_{F-MIG}^{ERSPs}\right]$, and $\left[PLV_{CG}^{ERSPs}\right]$ were the mean PLVs during MI-SW (0~10 s) in M-AIG, F-AIG, and CG, respectively. An edge was drawn between two brain regions only when the PLV value of the two brain regions was larger than the given threshold. In this way, the improved threshold interception method could not only effectively eliminate edges with weaker synchronization, but also was beneficial for between-group comparisons in the network analysis [38].

After the construction of brain networks, the differences between networks were analyzed by calculating the network parameters. The weight degree centrality of a node represented the sum of all weights of edges connecting to it. The eigenvector centrality [36] qualified the influence of a node in the network. The hubs owned high weight degree centrality, and their neighbor nodes also possessed high weight degree centrality. Hubs were the center of information as they could reach a large number of other nodes in the network via high weights and short paths. In this study, ERS/ERD values and eigenvector centrality were measured as qualified metrics for activation and functional connectivity strength of ROIs, respectively. Furthermore, hot spots were identified as the top 20% of strongly activated ROIs. Hubs were defined as top-level eigenvector centralities. We referred to previous studies where the number of hubs was consistent with hot spots [39]. When a node was both a hot spot and a hub, we called it a hot hub. Finally, the remaining nodes were defined as non-hub hot spots, cold hubs, and non-hub cold nodes.

#### 2.2.6. Statistical Analysis

Brainstorm [28] was used to calculate the ERS/ERD of 130 ROIs in the whole brain during MI-SW tasks, and the permutation T-test was used to compare the differences among M-AIG, F-AIG, and CG with a replace number of 10,000. A permutation T-test was not required for the data distribution and was particularly suitable for small sample data [40]. The false discovery rate (FDR) was applied to correct all results [41], with a threshold of *p* < 0.05. Finally, we calculated the effect size of ROIs’ activation using the Hedges’ g method applied to small samples [42].

## 3. Results

### 3.1. ERS Activation Mode

To evaluate the brain activation of AO-induced and non-induced MI-SW, we calculated the ERS/ERD for three groups. The spatial distributions of ERS/ERD signals in the α and β frequency bands are shown in Figure 2. During MI-SW, ERS was observed in a large number of brain regions during both AO-induced and non-induced tasks, with the strongest activation in bilateral occipital lobes.

### 3.2. Comparison of ROIs’ Activation between M-AIG and CG

We compared the differences in the activation of ROIs between AO-induced and non-induced tasks, in which the AO-induced tasks contained MI-SW elicited by males’ and females’ videos of swallowing. ROIs were colored according to the functional modules they belonging to in Figure 3 and Figure 4 (see Appendix A for the full name of the abbreviation). Figure 3 showed that compared with CG, the MI-SW of M-AIG elicited left-lateralized prominent activation including pars triangularis, precentral gyrus, pars opercularis, inferior occipital gyrus, right-lateralized remarkable activation including middle frontal gyrus, superior parietal gyrus, and bilateral superior temporal gyrus in alpha band (*p* < 0.05, FDR corrected). Furthermore, the supramarginal gyrus was inhibited.

Compared with CG, the ROIs significantly activated by M-AIG in beta band were in: (1) left-lateralized pars triangularis, superior parietal gyrus, pars orbitalis, supramarginal gyrus, cingulate gyrus, precuneus, subcallosal gyrus, fusiform gyrus, inferior temporal gyrus, and temporal pole; (2) right-lateralized cuneus, occipital gyrus, superior frontal gyrus, parahippocampal gyrus, and transverse temporal gyrus; and (3) bilateral angular gyrus, gyrus rectus, insula, middle frontal gyrus, orbito-frontal gyrus, paracentral lobule, postcentral gyrus, precentral gyrus, lingual gyrus, and middle temporal gyrus (see Figure 4, *p* < 0.05, FDR corrected). Almost all ROIs showed a medium (Hedges’ g > 0.5) or strong effect size (Hedges’ g > 0.8) [42].

### 3.3. Comparison of ROIs’ Activation between F-AIG and CG

Figure 3 showed that in the alpha band, F-AIG excited ROIs of left transverse temporal gyrus, insula, middle occipital gyrus, middle temporal gyrus, superior temporal gyrus, right pars triangularis, orbito-frontal gyrus, pars orbitalis, pars opercularis and bilateral middle frontal gyrus, superior parietal gyrus, and supramarginal gyrus, when compared with CG (*p* < 0.05, FDR corrected).

Furthermore, ROIs significantly activated by F-AIG in the beta band included: (1) left-lateralized insula, paracentral lobule, middle frontal gyrus, cuneus, middle occipital gyrus, and temporal pole; (2) right-lateralized pars opercularis, fusiform gyrus, and transverse temporal gyrus; and (3) bilateral angular gyrus, pars triangularis, orbito-frontal gyrus, transverse frontal gyrus, superior parietal gyrus, precuneus, superior frontal gyrus, middle temporal gyrus, and superior temporal gyrus (see Figure 5, *p* < 0.05, FDR corrected). All ROIs showed a medium (Hedges’ g > 0.5) or strong effect size (Hedges’ g > 0.8) [42].

### 3.4. Comparison of ROIs’ Activation between M-AIG and F-AIG

Figure 6 shows that compared with F-AIG, M-AIG elicited the activation of left pars triangularis in the alpha band. In addition, the left-lateralized remarkable activation included angular gyrus, postcentral gyrus, precentral gyrus, inferior occipital gyrus, and right paracentral lobule in the beta band (*p* < 0.05, FDR corrected). Furthermore, the left middle temporal gyrus, right supramarginal gyrus in alpha band, and left transverse temporal right pars opercularis in the beta band were inhibited. Almost all ROIs had medium (Hedges’ g > 0.5) or strong effects (Hedges’ g > 0.8) [42].

### 3.5. Comparison of Functional Brain Networks among Three Groups

Figure 7 reports the functional brain networks in alpha and beta frequency bands during the MI-SW of the three groups. The node size was proportional to ERSPs’ values in the figure. In the alpha band, the number of hot hubs (in red) in both CG and F-AIG was small, and they were distributed in the right occipital lobe, and there were no hot hubs in M-AIG. For non-hub hot spots (in yellow), which were mainly distributed in the right occipital lobe in CG, a bilateral occipital distribution was found in M-AIG and F-AIG. For cold hubs (in green), they were mainly distributed in the right prefrontal and limbic system in the CG, and in the bilateral prefrontal and limbic system in M-AIG and F-AIG.

In the beta band, there were no hot hubs in the CG and F-AIG, and the right gyrus rectus was invoked in the M-AIG. The distribution of non-hub hot spots in the three groups was similar, mostly concentrating in the bilateral occipital lobes. For cold hubs, they were mainly distributed in the left frontal and limbic lobes in the CG, and in the bilateral frontal and limbic lobes in the M-AIG and F-AIG.

## 4. Discussion

In limb movements, prior studies have suggested that compared to AO or MI alone, the combined application of AO and MI could enhance cortical excitability [43], motor performance [44], and brain functional connectivity [45]. In the case of swallowing, although it was a reflex movement, previous studies have confirmed the role of the cortical activity in swallowing [46,47,48,49,50,51,52]. Furthermore, Babaei et al. reported the presence of functional networks of swallowing-related cortical in healthy individuals [53], and the structural properties of cortical networks would affect the performance of swallowing. For example, postural compensation of the head and neck has been found to improve the efficiency of information transmission in brain networks [54], manifesting as improved swallowing performance and reduced risk of aspiration in patients [55,56]. These studies illustrated that the increased efficiency of information transfer in functional brain networks profited swallowing. In the present study, our major finding was that AO-induced MI-SW improved the activation level of swallowing-related ROIs and increased the number of non-hub hotspots and cold hubs in two hemispheres. Compared with CG, M-AIG and F-AIG showed a higher degree of swallowing-related ROIs’ activation. M-AIG and F-AIG increased functional connectivity between the bilateral hemispheres and improved the efficiency of information transmission in the brain network. In addition, our secondary finding was that there might be differences in the effects induced by viewing the stimulus material of different genders. The activation of swallowing-related ROIs was significantly stronger in M-AIG than in F-AIG.

### 4.1. ERS Pattern of MI-SW

Previous investigations on limb motor imagery have reported that short-term motor imagery could induce the ERD pattern in the contralateral brain cortex and ERS in the ipsilateral brain cortex, while long-term (5 s) motor imagery caused the ERS pattern [57,58]. Grazia et al. have reported that subjects performing 10 s of motor imagery of walking and bicycling after observing the movements showed an ERS pattern in the sensorimotor cortex, which was attributed to the fact that walking and cycling were closer to an automatic movement [33]. In this research, Figure 2 showed the spatial distribution of ERS/ERD of the whole brain: it showed that ERS predominated in nearly the whole brain in the alpha and beta bands in M-AIG, F-AIG, and CG. The probable reason for this finding could be the long-term swallowing imagery task and the spontaneous swallowing action that could occur in an unconscious state. Likewise, Yang et al. reported similar results that during the MI-SW (lasted 12 s) of healthy participants, a significant power increase (i.e., ERS) occurred in “C3” and “C4” regions in alpha and low-beta bands [21]. In addition, the ERS pattern was most significant in the occipital lobe, which might result from the subjects’ tendency to use visual imagery strategies. Since swallowing visual imagery was significantly higher than swallowing kinesthetic imagery scores in the KVIQ-10, swallowing kinesthetic imagery might be more difficult for subjects.

### 4.2. M-AIG and F-AIG versus CG: The Comparisons of Brain Activation

Kober et al. used near-infrared spectroscopy (NIRS) to detect hemodynamic changes in healthy young people during MI-SW, and the inferior frontal gyrus was the most significant [8,59]. In this study, AO-induced MI-SW significantly activated the pars opercular, pars triangular, and pars orbital in the inferior frontal gyrus. Additionally, a large number of swallowing-related ROIs were significantly activated in the M-AIG and F-AIG (Figure 3, Figure 4 and Figure 5), including the superior temporal gyrus, middle temporal gyrus, inferior temporal gyrus, transverse temporal gyrus, insula, and cingulate gyrus [7,49]. Brain regions associated with visual processing and facial recognition were also activated, e.g., the middle frontal gyrus, temporal pole, and fusiform gyrus [60]. In addition, ROIs of the ventral temporal association and the medial default mode system that encoded and retrieved episodic memory were also significantly activated [61] (Figure 4 and Figure 5). Activation of these visual function-related ROIs indicated the enhancement of visual imagery strategies. However, Jackson et al. argued that visual imagery primarily improved non-conscious processes related to movement and that the use of kinesthetic imagery might be more conducive to improved motor function [62]. Neuper et al. and Stinear et al. reported that more brain regions related to motor execution were activated by kinesthetic imagery than visual imagery [63,64]. Yang et al. found that the sensorimotor cortex was activated during the MI-SW using EEG [21]. In the present study, the sensorimotor cortex was significantly activated only in the M-AIG, suggesting that the AO contributing to the execution of kinesthetic imagery might be more conducive to the recovery of motor function. Furthermore, the attention and cinguloopercular system related to cognition and attention were significantly activated in the M-AIG and F-AIG [65] (Figure 3, Figure 4 and Figure 5). The invocation of cognitive- and attention-related ROIs might profit from the use of more difficult kinesthetic imagery strategies. Thus, the results suggested that both visual and kinesthetic imagery could be reinforced by AO induction and might make for higher training quality.

### 4.3. M-AIG and F-AIG versus CG: Functional Brain Network Connectivity Analysis

In functional brain networks, two types of data were included in each ROI, where the data for measuring the degree of brain activation belong to attribute data (i.e., ERS/ERD), and the network parameters were relational data. As shown in Figure 7, we constructed functional brain networks during the MI-SW under the three groups to investigate the differences in their activation and connection patterns. In order to exhibit these two kinds of attributes in the networks, we plotted “non-hub hot spots” (in yellow) with the high activation, “cold hubs” (in green) with the high connectivity, and “hot hubs” (in red) with both high activation and connectivity. In this study, the findings from functional brain networks (Figure 7) showed that non-hub hot spots and cold hubs were “separated”; that was to say, hot hubs were few. In patients with brain injury, “high-activation” regions tended to establish functional connectivity with the local-neighbor brain regions to recruit more neural resources when performing the same tasks [66]. Lin et al. reported that there will be more hot hubs in the injured brain, which would use more brain resources to complete the same tasks [39]. However, we found that in the present study, highly activated brain regions in healthy individuals tended to be weak in connectivity. This could be because healthy individuals were able to call brain resources more efficiently, and therefore adopted a different activation pattern. This finding could help in guiding the treatment of stroke patients and improving the invocation patterns of brain networks.

Furthermore, we found an effectiveness of AO induction on the invoking pattern of the swallowing functional brain networks. Non-hub hot spots (in yellow) in all three groups were mainly clustered in the occipital lobe. However, in the alpha band, unlike the right-sided distribution of CG, the number of non-hub hot spots in the left hemisphere was increased in the M-AIG and F-AIG. In addition, in both bands, the M-AIG and F-AIG increased the number of cold hubs in both hemispheres. Cold hubs could affect and transmit signals to multiple brain regions more efficiently. Therefore, we suggested that AO induction exerted beneficial effects on the invocation of the functional swallowing brain networks, which increased the activity of the relevant ROIs. The increasement of cold hubs in bilateral hemispheres might indicate the enhancement of the functional connectivity and the efficiency of information transmission between the hemispheres.

### 4.4. M-AIG versus F-AIG: The Comparisons of Brain Activation

Finally, we compared the induction effects of stimulus materials with different genders. A comparative analysis of data derived from watching swallowing videos of two genders revealed that a greater range and degree of activation were reported in the M-AIG (Figure 2). These additional activated brain regions of the M-AIG were located in the frontoparietal, sensorimotor, and ventral temporal associated systems (Figure 4), which were all associated with swallowing function [8,21]. Notably, compared to the CG, dominant activation was found in the bilateral postcentral gyrus and precentral gyrus, which were the most associated brain regions with swallowing movements, observed only in the M-AIG (Figure 3 and Figure 4). In addition, stronger activation of the M-AIG was observed in the sensorimotor cortex compared to the F-AIG (Figure 6). We speculated that this might be due to the fact that the up-and-down motion in AO materials, which only showed larynx close-ups, was the most important visual marker. In contrast to females, the larynx was more pronounced in males, and this might make it easier for subjects to focus on. This finding suggested that the AO of men swallowing might produce better training quality and effects. However, this secondary finding needs to be refined in future studies and clinical practice.

## 5. Limitations and Future Perspectives

The original major aim of this study was to investigate the effect of action observation induction on MI-SW. Further, during the analysis we additionally found a difference in the inductive effect of males’ and females’ videos on MI-SW and unfortunately, we were only able to make some hypothetical explanations for this secondary finding in the discussion and did not discuss it in detail. This was due to the fact that this issue required the subsequent design of a new experiment to verify the effect of various elements of the stimulus material on the results, such as the volume and luminance of the action observation videos, size and displacement of the laryngeal, and the gender of the viewer. In the future, we would conduct a detailed study of these factors that may affect the effectiveness of MI-SW. 

In addition, several limitations were noted in this study. First, the current study focused on the instant effect of a training session of MI-SW only, whereas the possible lasting effect of multiple training sessions was not considered. Second, although in this pilot study we used the permutation test and Hedges’ g to calculate the effect size which both applied to small sample studies, we realized that the sample size of this study was small. However, we referenced the previous studies of MI-SW in healthy individuals to selected the sample size [4,15]. Due to the subjects in this research were healthy young adults rather than brain injury patients, whose brain functional network recruitment patterns were more stable. The results of this study were able to initially support the AO-induced MI-SW training method. We would design a further experiment to further explore the effect of AO-induced MI-SW on post-stroke patients with dysphagia. The sample size would be further expanded in subsequent studies of patients with post-stroke dysphagia Ultimately, we would further explore whether functional brain networks of swallowing would be remodeled after long-term training (e.g., more connections between brain regions and more efficient in functional connectivity) and whether there were changes in behavioral during executing swallowing actions.

## 6. Conclusions

AO-induced MI-SW resulted in greater activation of swallowing-related brain regions, including visual, cognitive, attentional, and sensorimotor systems. AO induction also enhanced functional connectivity in the MI-SW bilateral hemispheres and reduced the number of hot hubs, thereby increasing resource utilization of brain networks. In addition, AO in males elicited greater activation of brain regions. The results of this a priori study confirmed the effectiveness and potential value of AO-induced MI-SW. It was expected to lay the foundation for further studies in patients with swallowing disorders in the future and provide evidence for the clinical application of AO-induced MI-SW in dysphagia rehabilitation.

## Figures and Tables

**Figure 1 brainsci-12-01420-f001:**
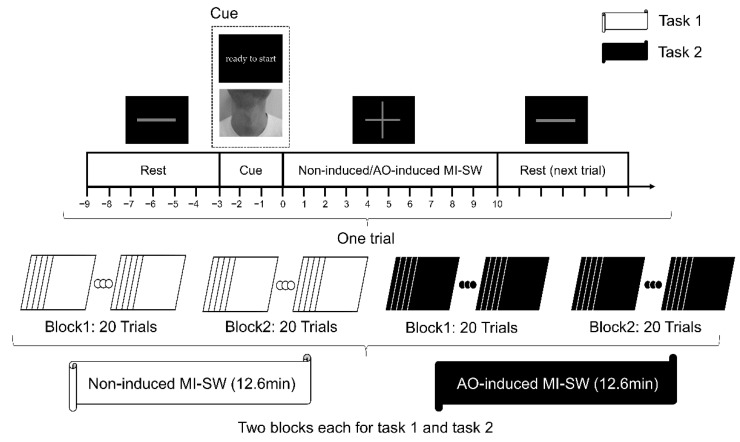
Flow chart for experimental paradigm. With the start of motor imagery of swallowing (MI-SW) task as the zero point, the baseline time window was −5~−3 s, the induction cue time window was −3~0 s, and the MI-SW time window was 0~10 s.

**Figure 2 brainsci-12-01420-f002:**
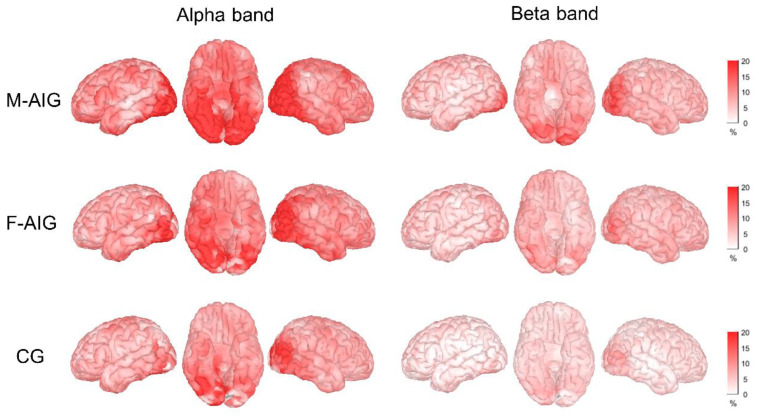
ERSPs map for the three groups in alpha and beta bands. The colors on the brain maps indicate ERS magnitudes (redder color indicates stronger ERS). M-AIG: male action observation-induced group; F-AIG: female action observation-induced group; CG: control group.

**Figure 3 brainsci-12-01420-f003:**
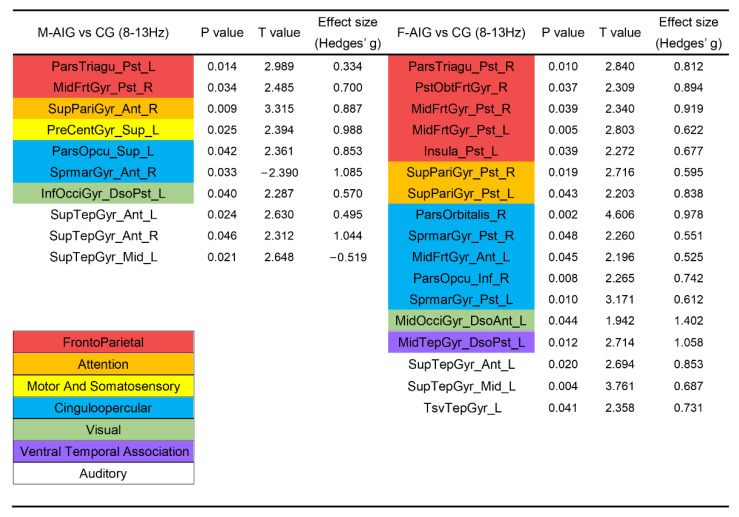
Comparison of ROIs’ activation between M-AIG, F-AIG, and CG in alpha band. M-AIG: male action observation-induced group; F-AIG: female action observation-induced group; CG: control group; L: left, R: right; *p* < 0.05, FDR corrected (for the full name of ROIs see Appendix A). Colors for functional modules: red: frontoparietal system; orange: attention system; yellow: motor and somatosensory system; blue: cinguloopercular system; green: visual system; purple: ventral temporal association; white: auditory system.

**Figure 4 brainsci-12-01420-f004:**
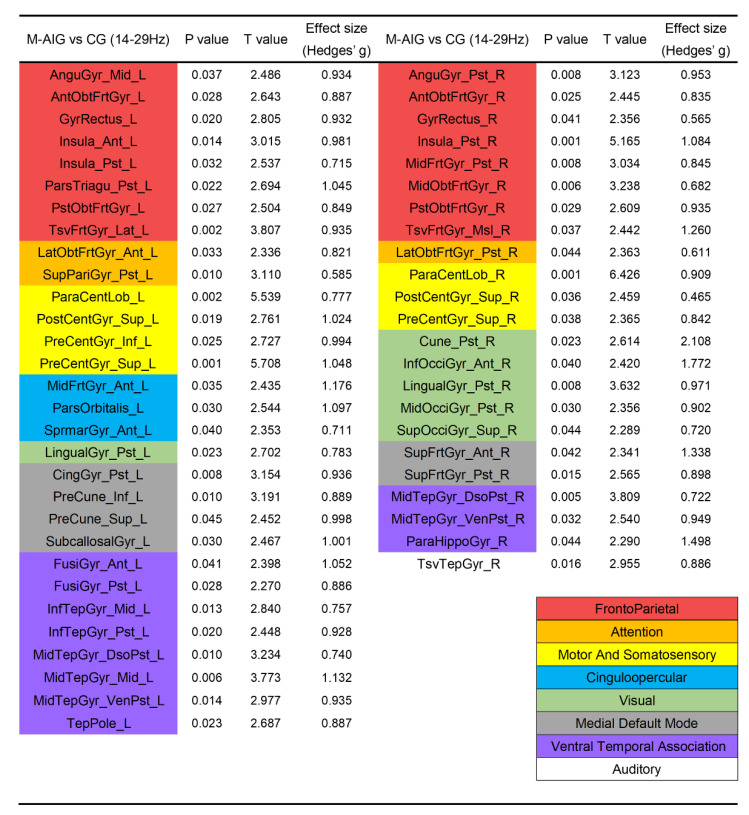
Comparison of ROIs’ activation between M-AIG and CG in beta band. M-AIG: male action observation-induced group; CG: control group; L: left, R: right; *p* < 0.05, FDR corrected (for the full name of ROIs see Appendix A). Colors for functional modules: red: frontoparietal system; orange: attention system; yellow: motor and somatosensory system; blue: cinguloopercular system; green: visual system; grey: medial default mode system; purple: ventral temporal association system; white: auditory system.

**Figure 5 brainsci-12-01420-f005:**
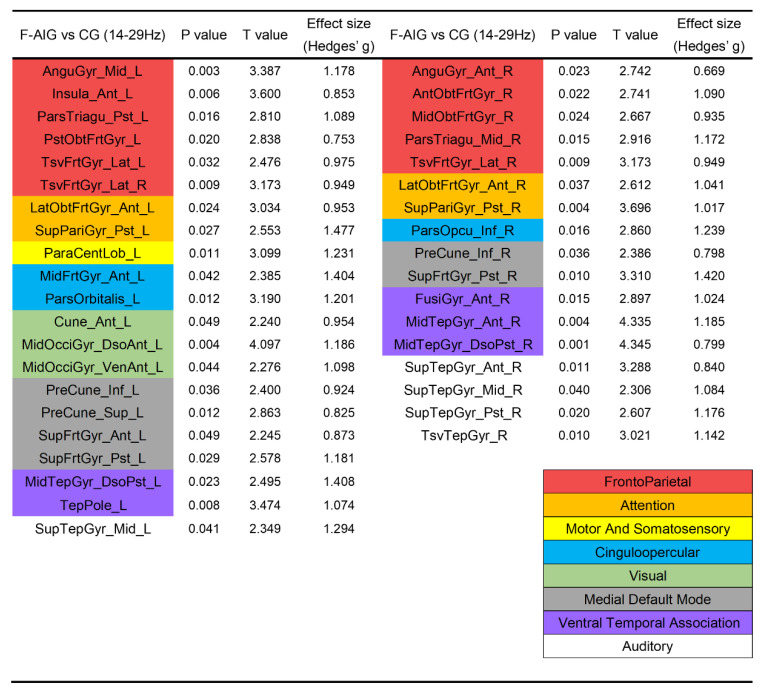
Comparison of ROIs’ activation between F-AIG and CG in beta band. F-AIG: female action observation-induced group; CG: control group; L: left, R: right; *p* < 0.05, FDR corrected (for the full name of ROIs see Appendix A). Colors for functional modules: red: frontoparietal system; orange: attention system; yellow: motor and somatosensory system; blue: cinguloopercular system; green: visual system; grey: medial default mode system; purple: ventral temporal association system; white: auditory system.

**Figure 6 brainsci-12-01420-f006:**
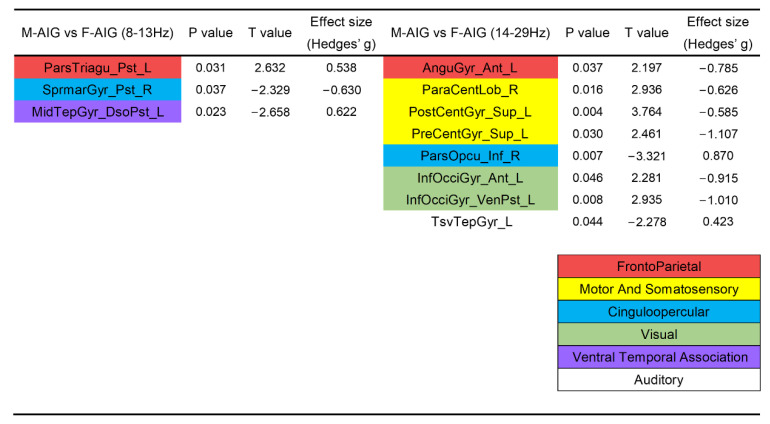
Comparison of ROIs’ activation between M-AIG and F-AIG in alpha and beta bands. M-AIG: male action observation-induced group; F-AIG: female action observation-induced group; L: left, R: right; *p* < 0.05, FDR corrected (for the full name of ROIs see Appendix A). Colors for functional modules: red: frontoparietal system; yellow: motor and somatosensory system; blue: cinguloopercular system; green: visual system; purple: ventral temporal association; white: auditory system.

**Figure 7 brainsci-12-01420-f007:**
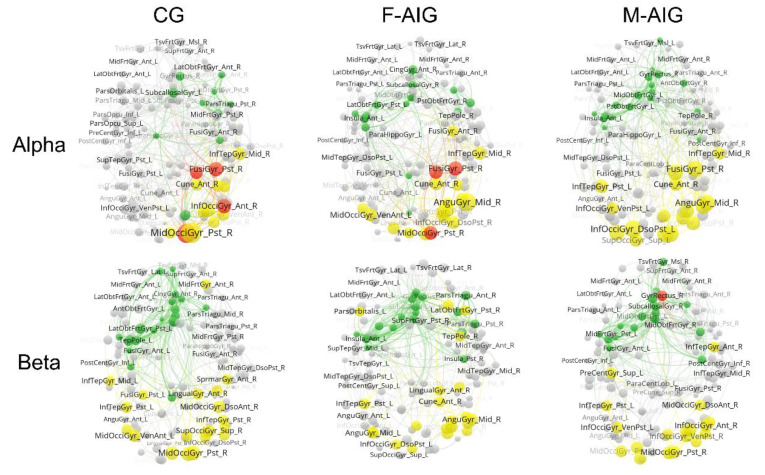
Distribution of hot hubs (in red), non-hub hot spots (in yellow), cold hubs (in green), and non-hub cold nodes (in grey) in brain function networks of three groups. All nodes are classified into four types: hot hubs (in red), non-hub hot spots (in yellow), cold hubs (in green), non-hub cold nodes (in grey). Node size is proportional to the ERSPs, and the view of the figure is top-down. M-AIG: male action observation-induced group; F-AIG: female action observation-induced group; CG: control group.

## Data Availability

Generated Statement: The raw data supporting the conclusions of this article will be made available by the authors, without undue reservation.

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
