# Peer review of "Activation Patterns of Functional Brain Network in Response to Action Observation-Induced and Non-Induced Motor Imagery of Swallowing: A Pilot Study"

_brainsci, 2022, doi:10.3390/brainsci12101420_

Round 1
Reviewer 1 Report
Reviewer’s Report on the manuscript entitled:
Activation Patterns of Functional Brain Network in Response to Action Observation-Induced and Non-induced Motor Imagery of Swallowing: A Pilot Study
The authors investigated the potential mechanisms for the action observation (AO) combined with motor imagery in swallowing. They showed that preliminary validation that AO induced swallowing imagery was more effective than swallowing imagery alone. Although the topic and the results are interesting, the presentation can be improved. Below, please see my comments.
Please insert the line numbers when you prepare the revisions.
Line 4 in abstract. Before “Twelve subjects performed” please say “In this study,”
First Line in Introduction. “Dysphagia was” or “Dysphagia is”?
At the end of Introduction, please mention how the rest of the manuscript is organized.
Section 2.1. First line. Please do not start a sentence with a number. Say here “Twelve” instead of “12”.
In Introduction, please define ROI.
Please define MRI. All abbreviations must be defined no matter how well-known they are.
Section 2.2.1. When you started the sentence with EEG, please say Electroencephalogram (EEG) and add the following references which describes its applications and limitations:
https://doi.org/10.3390/s22062346
https://doi.org/10.3390/signals3030035
Figure 7, Caption. It should be “three groups”. Also, please mention what the red, green, yellow, and grey colors mean in the caption of Figure 7.
In the bottom of page 12, it should be “Lin et al.” not “Lin et. al”
The conclusion section is very short. Please mention the limitations of this study and future direction in the Conclusion section. Basically, you can merge Sections 5 and 6 into a section called “Conclusions and Future Direction.” At the beginning of this section, please mention the objective of this study and then talk about the findings, then limitation, then future direction.
Please follow the MDPI guidelines for formatting the references.
Thank you for your contribution
Regards,
Author Response
Thank you very much for your suggestions, we have responded to your comments and revised the manuscript. Please see them in the attachment.

Reviewer 2 Report
While I have no concerns on the methodological part of the work by Lin et al, I would enrich the work by more elaboration on the possible clinical implications. In the introduction authors state:
"Dysphagia was a common post-stroke dysfunction that more than 70% of stroke sur- vivors have dysphagia or difficulty swallowing [1]. Compared to conventional rehabilita- tion treatment programs, safe treatment methods, such as motor imagery (MI) and action observation (AO) were developed in rehabilitation of neurological disorders. AO and MI could improve motor function and performance by activating mirror neurons similar to those activated by actual movemen"
I believe that it would be valuable to mention and elaborate on other diseases affected by swallowing impairments e.g. neurodegenerative:
Management of dysphagia and gastroparesis in Parkinson's disease in real-world clinical practice - Balancing pharmacological and non-pharmacological approaches. Front Aging Neurosci. 2022 Aug 11;14:979826. doi: 10.3389/fnagi.2022.979826. PMID: 36034128; PMCID: PMC9403060.
Progressive Supranuclear Palsy-Parkinsonism Predominant (PSP-P)-A Clinical Challenge at the Boundaries of PSP and Parkinson's Disease (PD). Front Neurol. 2020 Mar 10;11:180. doi: 10.3389/fneur.2020.00180. PMID: 32218768; PMCID: PMC7078665.
It would be valuable to discuss other neuroimaging options of swallowing
Author Response
Thank you very much for your suggestion, we have responded to your comments and revised the manuscript. Please see it in the attachment.

Reviewer 3 Report
Thank you for permitting me to review this manuscript
In this study 3 groups of subject were compared control , male and female AO
They found that male and female AO induced group significantly activated sensorimotor cortex compared to control groupM-AIG significantly activated sensorimotor cortex compared to CG and F-AIG.
the abstract need to be rephrased in order to better understand the whole protocol and results
methods
male and women separation are not reaally explained , I do not see any rationale for this please insert additional justification to explain this split
The sample size as strated by the authors themselves is small
Discussion
In the present study, our major finding was that AO-induced MI-SW improved the range and efficiency of invocation of the functional brain network of swallowing
This appears to be more speculative than fact related
in addition young adlts did'nt have swaowing defects therefore how could we be sure that this network affected swallowing
Therefore, we suggested that AO induction exerted beneficial effects on the invocation of the functional swallowing brain networks, which not only increased the activity of the relevant ROIs, but also enhanced the functional connectivity and the efficiency of information transmission between the two hemispheres.
I think again this latter suggestion is mainly speculative please adapt statement related to the facts
Therefore, we suggested that AO induction exerted beneficial effects on the invocation of the functional swallowing brain networks, which not only increased the activity of the relevant ROIs, but also enhanced the functional connectivity and the efficiency of information transmission between the two hemispheres.
Please elborate this statement above
Limitations and future perspectives
it is stated that the difference between males and females were found significant secondly but how come this split was performed in the method section
Author Response

(The authors gave the same response as above.)

Reviewer 4 Report
This is a very interesting paper investigating patterns of functional brain network. The paper is well-written and of interest for the readers; however, several minor changes should be made before considering it for publication.
Abstract.
1- I recommend to add the "aims or objectives" in the abstract section. It is necessary to understand why the authors are investigating action-observation induced and non-induced motor imagery.
2- The methods used should be summarized. In the abstract the authors are only reporting (in the methods) the three groups of comparison.
Introduction.
1- The first sentence is repeatingthat dysphagia is frequent in stroke survivors. Please, rephrase it.
2- I recommend to describe other complications of stroke related to respiratory or gastrointestinal function.
3- The main aims of the study should be described in a separate subsection. I suggest to add a "1.1. Section called "Aims".
Methods:
1- At the beginning of a sentence numbers are not recommended. Please, rephrase the first sentence. "Twelve...".
Results
1- The subsections of the results section are describing different analyses. I recommend to rename the sections according to the comparisons between the three groups. It's just a suggestion.
2- Gender comparisons were made by the authors. These analyses should be grouped in a separate section to highlight the results.
Discussion.
1- The discussion is divided according to the main results. I considerat that it has no sense to divide the section into several subsections repeating the structure.
2- The discussion should be focused on the main results, and discussing the findings with other studies in the field...
Author Response

(The authors gave the same response as above.)

Round 2
Reviewer 2 Report
Thank you for the corrections, however I believe that an additional feature mentioned in the first round should be extended
"It would be valuable to discuss other neuroimaging options of swallowing"
Moreover in my opinion authors should refer to various disorders related with swallowing disturbance in the introduction, not only to PD and stroke.
Author Response
We are sorry that we misunderstood your meaning, we have added the content about other diseases causing swallowing disorders, which was revised in the manuscript as: “Dysphagia is a common dysfunction in neurological disorders, which commonly occurred in stroke, Parkinson's disease, brain injury, dementia, multiple sclerosis, amyotrophic lateral sclerosis and cerebral palsy [1–3]. In addition, impairment of swallowing-related organic structures (e.g. neck and oropharyngeal cancer) and aging could lead to dysphagia [1,3]. Dysphagia is associated with nutritional and respiratory impairment, and increases risk of aspiration, mortality and medical expenses [1,4]. Compared to conventional rehabilitation programs of neurological disorders, motor imagery (MI) and action observation (AO) which were developed in recent years avoided risks of aspiration [5]. “
Due to the large variety of diseases mentioned, in order to reduce the number of literature citations, we cited two reviews that reported on these diseases that cause swallowing disorders.
Reviewer 3 Report
The authors have significantly improved the manuscript
Author Response
We sincerely thank you for your helpful suggestions which help us improve our manuscript. Best regards!